# Tapered whiskers are required for active tactile sensation

Samuel Andrew Hires[1], Lorenz Pammer[1,2], Karel Svoboda[1]*, David Golomb[1,3,4]*

[1]Janelia Farm Research Campus, Howard Hughes Medical Institute, Ashburn, United States; [2]Max Planck Institute for Brain Research, Frankfurt am Main, Germany; [3]Department of Physiology and Cell Biology, Ben Gurion University, Be'er-Sheva, Israel; [4]Zlotowski Center for Neuroscience, Ben Gurion University, Be'er-Sheva, Israel

**Abstract** Many mammals forage and burrow in dark constrained spaces. Touch through facial whiskers is important during these activities, but the close quarters makes whisker deployment challenging. The diverse shapes of facial whiskers reflect distinct ecological niches. Rodent whiskers are conical, often with a remarkably linear taper. Here we use theoretical and experimental methods to analyze interactions of mouse whiskers with objects. When pushed into objects, conical whiskers suddenly slip at a critical angle. In contrast, cylindrical whiskers do not slip for biologically plausible movements. Conical whiskers sweep across objects and textures in characteristic sequences of brief sticks and slips, which provide information about the tactile world. In contrast, cylindrical whiskers stick and remain stuck, even when sweeping across fine textures. Thus the conical whisker structure is adaptive for sensor mobility in constrained environments and in feature extraction during active haptic exploration of objects and surfaces.

## Introduction

Many mammals use facial whiskers for navigation (*Vincent, 1912*; *Dehnhardt et al., 2001*), object localization (*Hutson and Masterton, 1986*; *Knutsen et al., 2006*; *Krupa et al., 2001*; *Mehta et al., 2007*; *O'Connor et al., 2010a*; *Pammer et al., 2013*), texture discrimination (*Carvell and Simons, 1990*; *Wolfe et al., 2008*; *Chen et al., 2013*), and object recognition (*Anjum et al., 2006*). The shapes of mammalian whiskers are diverse. Rodent whiskers are conical (*Birdwell et al., 2007*; *Williams and Kramer, 2010*; *Quist et al., 2011*; *Pammer et al., 2013*), whereas sea lion whiskers (*Hanke et al., 2010*) and human hair are approximately cylindrical. Whiskers of harbor seals have elliptical cross-sections with an undulated structure (*Hanke et al., 2010*). Differences in whisker shapes across different species likely reflect differences in how animals use their whiskers. For example, the undulating microstructure of harbor seal whiskers suppresses vibrations triggered by vortices and enhances the seal's ability to analyze water movements (*Hanke et al., 2010*).

What could be the advantages of the whisker taper seen in rodents? Rodents sense their surroundings by moving their whiskers over objects with large amplitudes (up to 50° peak–peak) in a rhythmic motion (*Knutsen et al., 2006*; *O'Connor et al., 2010a*; *O'Connor et al., 2013*; *Voigts et al., 2008*). Rodents can localize and recognize objects in three dimensions (*Knutsen et al., 2006*; *Krupa et al., 2001*; *O'Connor et al., 2010a*; *Pammer et al., 2013*; *Voigts et al., 2008*) and also discriminate subtle differences in surface textures (*Carvell and Simons, 1990*; *Wolfe et al., 2008*) (reviewed in *Diamond, 2010*). These behaviors are based on collisions between whiskers and objects, which cause time-varying forces at the whisker base and excitation of sensory neurons in the follicles (*Zucker and Welker, 1969*; *Szwed et al., 2006*). Whisker mechanics thus couples the tactile world to forces at the whisker base (*Solomon and Hartmann, 2006*; *Birdwell et al., 2007*; *Bagdasarian et al., 2013*; *Pammer et al., 2013*).

*For correspondence:
svobodak@janelia.hhmi.org (KS);
golomb@bgu.ac.il (DG)

**Reviewing editor**: Misha Tsodyks, Weizmann Institute of Science, Israel

**eLife digest** When foraging in dark, confined spaces, mammals use the information gathered by their whiskers to 'see' the world around them. Mammalian whiskers come in a variety of shapes and sizes, most likely reflecting the way in which they are used. Rodent whiskers are conical and precisely tapered, whereas some harbor seals have flattened whiskers with wave-like undulations. Human hair is cylindrical.

Rodents sweep their whiskers back and forth over objects and surfaces without moving their head. They use this process, called whisking, to build up a three-dimensional picture of objects. Whisking allows the rodent to estimate where an object is located, how big it is, and what kind of surface texture it has. Information about surface texture can, for example, help the animal to distinguish a stone from a seed.

Hires et al. have used theoretical and experimental methods to analyze the interaction of mouse whiskers with objects. The conical shape of a mouse whisker makes the tip thousands of times more flexible than the base. Hires et al. show that this flexibility gradient allows the whiskers to slip past objects close to the face and to move freely across rough surfaces. Cylindrical whiskers, on the other hand, become stuck behind nearby objects and get caught on tiny features in an object's surface texture.

Hires et al. conclude that conical whiskers are advantageous in the tight confines of the tunnels that mice live, forage and socialize in, because they are able to gather a more complete sensory picture of their surroundings. The maneuverability of the whiskers also allows the mouse to move their whiskers forwards or backwards when rough tunnel walls are close by. By contrast, the sticking experienced by cylindrical whiskers would lead to 'blind spots'. In addition to providing insights into the ways that mice interact with their environment, this work could also lead to improvements in the design of the canes used by the visually impaired to navigate human environments.

Rodent whiskers are thin, approximately linear and homogenous elastic cones (*Solomon and Hartmann, 2006*; *Birdwell et al., 2007*; *Williams and Kramer, 2010*; *Pammer et al., 2013*). As a result of the linear taper, whisker bending stiffness decreases with distance from the face over five orders of magnitude. Behavioral measurements have shown that mice use distance-dependent whisker mechanics as a ruler to estimate object location along the length of the whisker (*Pammer et al., 2013*).

Here we used theoretical and experimental methods to analyze the interactions of whiskers with objects. We uncover additional decisive advantages of conical whiskers compared to cylindrical whiskers for tactile exploration. Conical whiskers sweep across textures with informative micromotions, whereas cylindrical whiskers get stuck. The steep increase in flexibility from base to tip of conical whiskers allow rodents to maneuver their sensors past objects with relative ease. Conical whisker shape is thus critical for tactile exploration in confined spaces.

## Results

### Mechanical model of whiskers interacting with an object

We modeled rodent whiskers as truncated cones with a cylindrical cross section, base radius $r_{base}$, tip radius $r_{tip}$, and length $L_W$ (*Ibrahim and Wright, 1975*; *Boubenec et al., 2012*) (*Figure 1A*). Whiskers have intrinsic curvature (*Quist and Hartmann, 2012*) and are further deflected by forces that are caused by interactions with objects (*O'Connor et al., 2010a*; *Bagdasarian et al., 2013*; *Pammer et al., 2013*) (*Figure 1B*). In our model, contacts occurred either in the 'concave backward' (CB) or 'concave forward' (CF) directions (*Figure 1C*) (*Quist and Hartmann, 2012*). We quantified contact strength using the push angle $\theta_p$ (*Quist and Hartmann, 2012*), the angle through which the whisker is rotated into the object (*Figure 1D*). By convention, contacts for the CB configuration correspond to $\theta_p > 0$, and for the CF configuration to $\theta_p < 0$; $\theta_p = 0$ defines the angle of initial touch. In all cases whisker movement and bending were limited to the $x$-$y$ plane. We computed whisker shape by solving the Euler–Bernoulli beam equation in the quasi-static regime (*Euler, 1744*; *Birdwell et al., 2007*; *Solomon and Hartmann, 2006*; *Pammer et al., 2013*). The beam equation describing whisker shape was converted to a boundary-value problem formulation ('Materials and methods'; *Press et al., 1992*), a set of differential equations

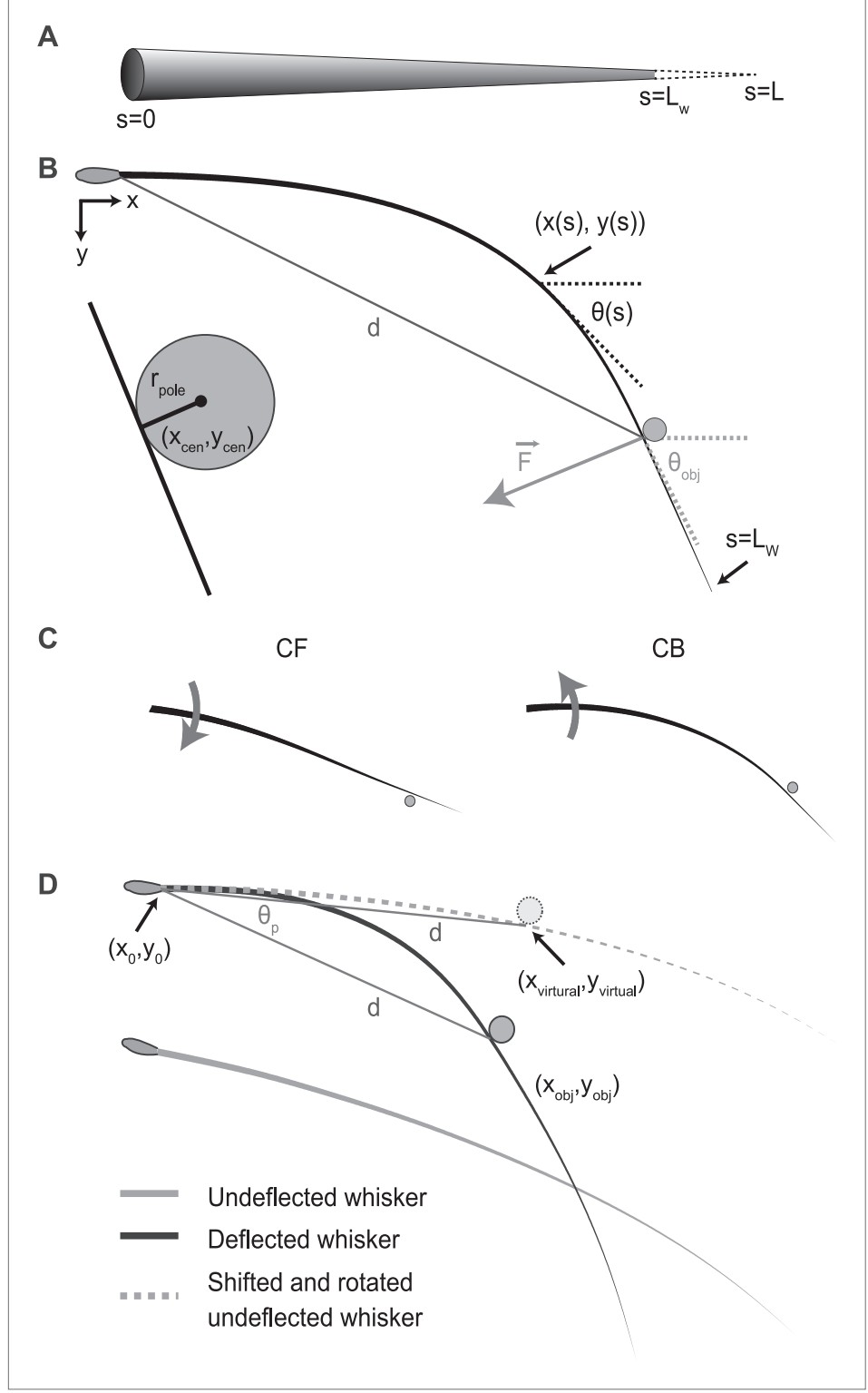

**Figure 1**. Schematic of the whisker in two dimensions. (**A**) The whisker is modeled as a truncated cone of length $L_W$, virtually extended to length $L$. (**B**) The base of the whisker (in the follicle, or attached to a galvo, **Figures 4–6**) is at point $(x_0, y_0)$ and angle $\theta_0$, measured clockwise. The position of a point along the whisker is $(x(s), y(s))$ and its angle with the x-axis is $\theta(s)$. The contacted object is a cylindrical pole with radius $r_{pole}$ centered at $(x_{cen}, y_{cen})$; the pole and whisker are shown at a magnified scale in the inset on the left. The whisker contacts the object at the point $(x_{obj}, y_{obj})$

*Figure 1. Continued on next page*

*Figure 1. Continued*

at an angle $\theta_{obj}$. The object distance, *d*, is the distance between ($x_{obj}$, $y_{obj}$) and the whisker base. The pole applies a force *F* on the whisker. (**C**) The concave forward (CF, left) and concave backward (CB, right) whisker configurations. Thick black lines, whiskers; solid circle, poles; gray arrows, movement directions. (**D**), Definition of the push angle, $\theta_p$, which measures the strain on the whisker imposed by the object. The deflected and undeflected whiskers are shown as black and gray lines respectively. The pole is a dark gray circle. The undeflected whisker is translated and rotated in the plane such that it has the same $x_0$, $y_0$ and $\theta_0$ as the deflected whisker. This generates a virtual undeflected whisker (dashed gray line). A virtual pole (light gray circle) is generated by shifting the real pole such that it will be tangent to the virtual undeflected whisker. In addition, the distance from the contact point of the virtual unbent whisker and the virtual pole, ($x_{virtual}$, $y_{virtual}$) and the base ($x_0$, $y_0$) is equal to *d*, the distance between ($x_{obj}$, $y_{obj}$) and the base. The angle between the two line segments connecting the base with the real and virtual contact points is $\theta_p$.

with defined boundary conditions at the whisker base and at the point of contact with the object. The object was assumed to be a cylindrical pole perpendicular to the plane of motion, as is typically used in object localization experiments (***Knutsen et al., 2006***; ***Mehta et al., 2007***; ***O'Connor et al., 2010a***; ***Pammer et al., 2013***) (***Figure 1B***). The whisker shape at each time was determined by the static solution computed for the time-varying boundary conditions. Using identical methods we also modeled hypothetical cylindrical whiskers.

The boundary-value problem for whisker shape generally has two solutions, one stable and the other unstable (***Figure 2A***). During object contact, the whisker shape matches the stable solution since small perturbations from it will decay back to the stable solution (***Strogatz, 1994***). The bending of the stable solution is weaker compared to the unstable solution. As the whisker pushes further into the object it becomes increasingly deflected (***Figure 2B***). At the same time the whisker slides along the object and the arclength from the whisker base to the point of contact, $s_{obj}$, increases until the whisker detaches from the object. We note two qualitatively different types of detachment. First, under some conditions detachment occurs suddenly before the end of the whisker has reached the object, $s_{obj} < L_W$; we refer to this type of detachment as 'slip-off' (***Figure 2B***). Second, detachment has to occur when the tip reaches the end of the object, $s_{obj} = L_W$; we refer to this type of detachment as 'pull-off' (***Figure 2C,D***).

Bending of the whisker can be characterized by the angle of the whisker at the point of object contact, $\theta_{obj}$ (***Figure 1B***). The whisker first touches the pole at $\theta_p = 0$ (***Figure 3A–D***, open circles). As the whisker pushes into the pole ($|\theta_p| > 0$), $\theta_{obj}$ changes monotonically (***Figure 3B***). In the CB configuration, the whisker bends and its shape becomes more 'concave backwards'. The force *F* acting on the whisker increases as more elastic energy is stored in the whisker (***Figure 3C***); $s_{obj}$, also increases (***Figure 3D***). At a critical $\theta_p$, the two solutions (solid lines, dashed lines) coalesce and disappear at a saddle-node bifurcation (SNB) (***Strogatz, 1994***) (***Figure 3B–D***, solid circles). No solution exists above this critical $\theta_p$ value, which also corresponds to a critical $s_{obj} < L_W$. The whisker slips suddenly and rapidly past the pole. The saddle-node bifurcation corresponds to slip-off. In the CF configuration, $s_{obj}$ first decreases as $|\theta_p|$ increases from 0, because touch forces straighten the whisker (***Figure 3D***). With further increases in $\theta_p$, the whisker bends in the other direction and $s_{obj}$ increases.

The regime in which a stable static solution exists for whisker shape can be visualized by plotting a 'detachment' curve in the $\theta_p - d$ plane, where *d* is the distance of the object from the base of the whisker (***Figure 3E***). The detachment curve is the set of points in the $\theta_p - d$ plane where detachments occur. It encloses an area where stable contacting solutions exist. When the object is close to the face (small *d*), the whisker contacts the object near its base and a static solution exists for most practical $\theta_p$ values (peak-to-peak amplitude of whisker movements, 50° [***Voigts et al., 2008***; ***Curtis and Kleinfeld, 2009***; ***O'Connor et al., 2013***]). For larger *d*, the $\theta_p$ regime with a stable solution decreases approximately linearly. Detachments correspond to slip-offs. The range of $\theta_p$ with a stable solution is larger for the CF than the CB configuration. This is consistent with experimental observations (***Quist and Hartmann, 2012***) and intuition: in the CB configuration the intrinsic curvature aids slip-off. When the object touches near the whisker tip (large *d*), the saddle-node bifurcation ceases to exist and the whisker is 'pulled off' the object. (***Figure 3E***, blue line). The pull-offs are the result of whisker truncation, and would not occur for a perfect cone.

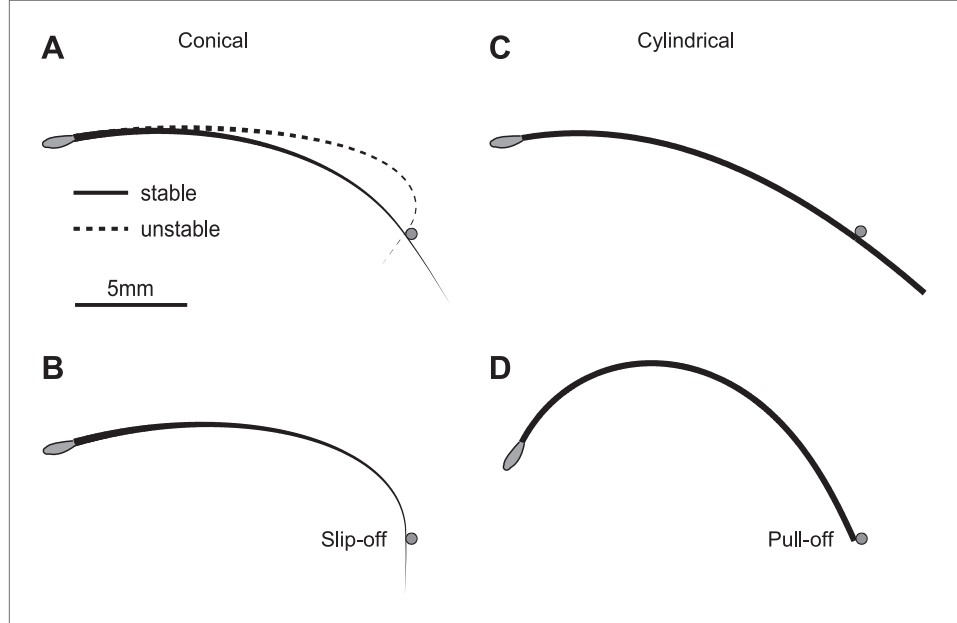

**Figure 2**. Interactions between whiskers and an object. Solutions of the quasi-static model (*Equations 8–14*) for a conical whisker (**A** and **B**) and a cylindrical whisker (**C** and **D**). The pole is denoted by a gray circle. The resting shape of the whisker is $y = Ax^2$, where $A = 0.02$ mm$^{-1}$ (*Quist and Hartmann, 2012*), and the whisker touches the pole in the concave backward configuration. (**A**) Two solutions for a conical whisker. For $\theta_p = 10°$, there are two solutions for whisker shape, one is stable (solid line) and one is unstable (dashed line). (**B**) Whisker shape at the saddle-node bifurcation ($\theta_p = 15.6°$). There is only one solution as the stable and unstable solutions coalesce. The object touches the whisker not at the tip. For any larger $\theta_p$, static solutions cease to exist and the whisker slips off the pole. (**C**) Whisker shape for a cylindrical 'whisker' and $\theta_p = 10°$. Only one solution (stable) exists; the unstable solution is not physical because its computed arclength is longer than $L_w = 20$ mm. (**D**) For $\theta_p = 62.7°$, the tip of the cylindrical 'whisker' reaches the object. Beyond this value of $\theta_p$, the 'whisker' is pulled off the object. Parameters for all panels: $L_w = 20$ mm, $r_{base} = 30$ µm, $r_{tip} = 1.5$ µm for the conical whisker and 30 µm for the cylindrical 'whisker', $d=15.7$ mm, $E = 3$ GPa, $x_0 = 0$, $y_0 = 0$. The pole has $r_{pole}=0.25$ mm and its center is located at $x_{cen} = 15.13$ mm, $y_{cen} = 4.29$ mm.

Identical analyses were performed for hypothetical cylindrical whiskers (*Figure 3F–J*). Although the bifurcation diagrams were superficially similar for conical and cylindrical whiskers (c.f. *Figure 3B–E,G–J*), cylindrical whiskers exhibit stable solutions at much larger $\theta_p$. The SNB occurs for $\theta_p > 90°$ (*Figure 3G–I*, black lines and solid circles; $d = 10$ mm), which is beyond plausible ranges of whisking since whiskers cannot move into the face. When cylindrical whiskers touch the object close to their end, they are pulled off at moderate $\theta_p$, because the whisker tip reaches the object (*Figure 3I*, blue line) $s_{obj} = L_W$. Therefore, cylindrical whiskers do not slip-off the pole. For a homogenous cylinder this effect is independent of the cylinder's bending stiffness and thus its thickness. Our model thus predicts that conical and cylindrical whiskers interact with objects in a fundamentally different way. For a large range of object distances conical whiskers slip past objects, whereas cylindrical whiskers get stuck. This difference is expected to have profound consequences for object-whisker interactions during haptic sensation.

## Whisker-object interactions

We compared our model with measurements made on mouse whiskers (conical) and human hair (cylindrical) (*Figure 4*). Mouse and human hair have similar Young's Modulus (*Hu et al., 2010*; *Quist et al., 2011*). A C2 mouse whisker was mounted on a galvanometer scanner so that its intrinsic curvature was in the plane of whisker movement (*Figure 4A*). The whisker was then moved slowly ($f_{galvo} = 0.2$ Hz, peak-to-peak amplitude 30°) against a pole. As the whisker rotated into the pole it was deformed until, at a critical $\theta_p$, it suddenly slipped off the pole (*Figure 4B*, *Video 1*). The red line in *Figure 4B* shows the whisker immediately before detachment. Whisker slip-offs occurred before the tip of the whisker had reached the point of contact. In contrast, for the cylindrical hair, slip-offs did not occur. Detachments

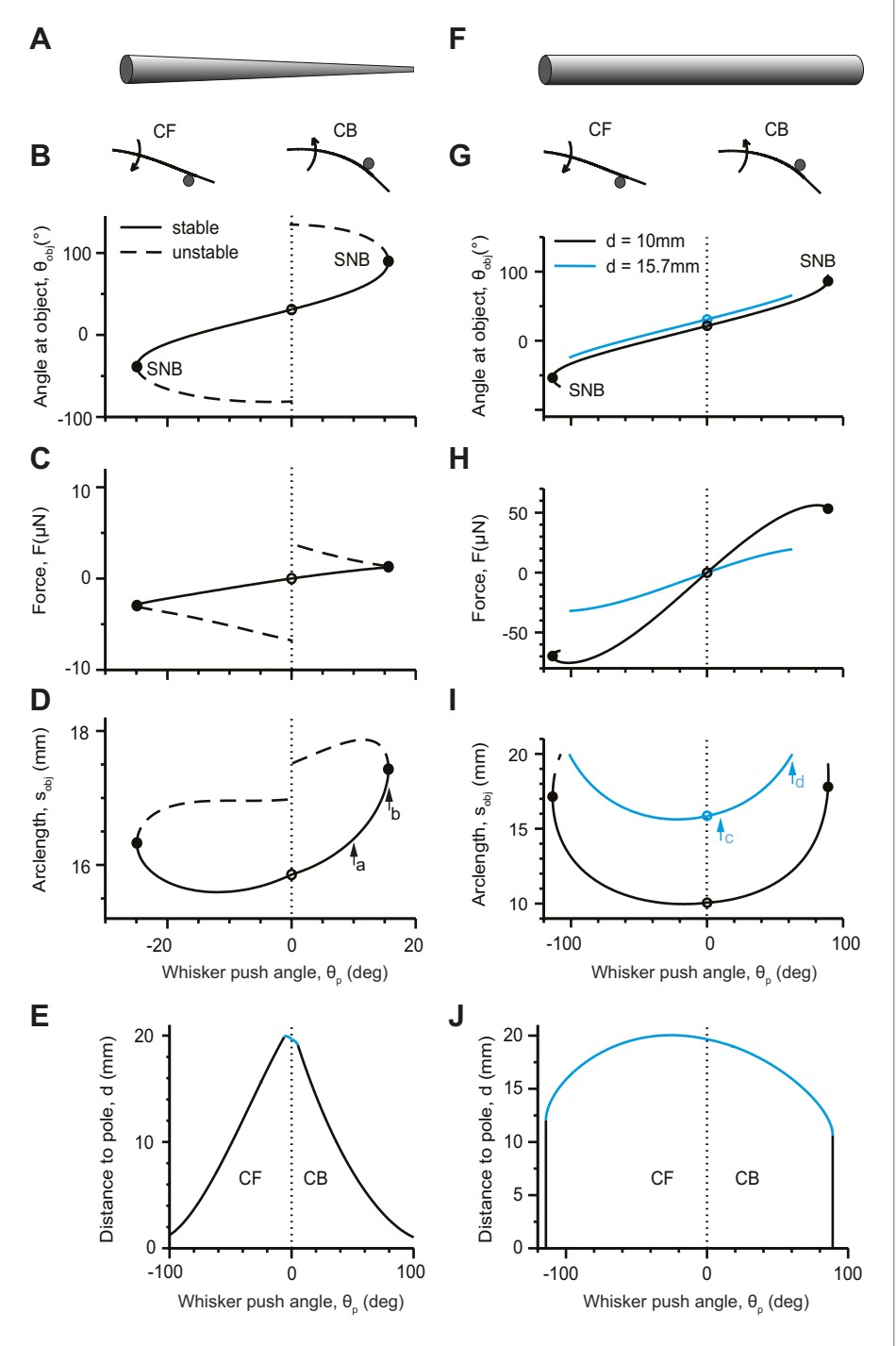

**Figure 3**. Analysis of conical (**A**–**E**) and cylindrical whiskers (**F**–**J**) pushing into a pole. (**A**) Schematic of a conical whisker. Parameters for panels (**B**–**E**): $L_w$ = 20 mm, $r_{base}$ = 30 µm, $r_{tip}$ = 1.5 µm, $x_0$ = 0, $y_0$ = 0, $r_{pole}$ = 0.25 mm, $E$ = 3 GPa. The equation of the undeflected whisker is $y = Ax^2$ where A = 0.02 mm$^{-1}$ (***Quist and Hartmann, 2012***). For CB configurations, $x_{cen}$ = 15.13 mm, $y_{cen}$ = 4.29 mm; for CF configurations, $x_{cen}$ = 14.87 mm, $y_{cen}$ = 4.71 mm. Positive and negative values of $\theta_p$ correspond to CB and CF configurations respectively. (**B**) $\theta_{obj}$ as a function of $\theta_p$. Left, concave forward (CF); right, concave backward (CB). Solid lines, stable solutions; dashed lines, unstable solutions (***Equations 8–14***). Solid circles denote saddle-node bifurcations (SNB). (**C**) Force $F$ as a function of $\theta_p$. (**D**) Location of object contact along the whisker arc, $s_{obj}$, as a function of $\theta_p$. Arrows correspond to ***Figure 2A*** (a) and ***Figure 2B*** (b, SNB). (**E**) The detachment curve in the $\theta_p$–$d$ plane bounds the parameter regime with a stable solution for a

*Figure 3. Continued on next page*

*Figure 3. Continued*

whisker contacting an object. Black lines represent the points when the stable solution coalesces with an unstable solution and disappears via a saddle-node bifurcation (slip-offs). Blue line represents the points where the whisker is pulled off because the tip has reached the object, $s_{obj} = L_W$ (pull-offs). (**F**) Schematic of a cylindrical whisker. Parameters as for conical whisker, except that $L_w = 20$ mm, $r_{base} = r_{tip} = 30$ µm. Panels (**G**–**J**) correspond to panels (**B**–**E**). Two object distances are considered in panels (**G**–**I**). Arrows in panel **i** correspond to *Figure 2C,D*. $d = 15.7$ mm (blue lines) corresponds to the pole location used in (**B**–**D**). The ends of the blue lines correspond to pull-offs. Additionally, an object distance $d = 10$ mm is shown (black lines). The black solid circles correspond to slip-offs (SNBs).

always coincided with the whisker tip reaching the point of contact and were thus pull-offs (*Figure 4C*, red line, *Video 2*).

We performed the same type of measurement for multiple object locations along the whisker (*d*, *Figure 4A*). The regime of stable interactions between whisker and pole, bounded by the detachment curve, can be visualized in the $\theta_p-d$ plane (*Figure 4D*). The experimental results were in agreement with the model. For conical whiskers, slip-off occurred before the whisker tip reached the object, and the critical $\theta_p$ decreased rapidly with object distance (*Figure 4D*, black circles). The observed deviations between the idealized conical model and actual whisker are expected because the whisker is not a perfect cone (*Ibrahim and Wright, 1975*) and because the whisker's Young's modulus may vary slightly along its length (*Quist et al., 2011*). In contrast, the cylindrical hair only pulled off when the whisker tip reached the pole (*Figure 4C*), with a close fit between experimental and theoretical results (*Figure 4D*, blue circles).

We next tested if slip-offs occur normally during whisker-dependent behavior (*Figure 5*). We analyzed data from head-fixed mice trained in a vibrissa-based object location discrimination task (*Pammer et al., 2013*). Mice reported the presence of a pole at a target position (the 'Go stimulus'; proximal) or in a distracter position (the 'No Go stimulus'; distal) (*Figure 5A*) by either licking (Go response) or withholding licking (No Go response). In each trial, the pole was presented at a single location. Whiskers were trimmed so that mice performed the task with a single whisker (C2). For the trials analyzed here the pole distance from the face was randomly chosen from the range d = 7–13 mm (measured from the follicle; the No Go stimuli). We used high-speed (500 Hz) videography and automated whisker tracking to measure the position and shape of the whisker in two mice (*Clack et al., 2012*; *Pammer et al., 2013*) (140 slip events).

In behaving mice, the intrinsic whisker curvature is not parallel to the plane of whisking and imaging (*Towal et al., 2011*). We corrected for the curvature out of the imaging plane using a simple procedure ('Materials and methods'). Furthermore, whiskers exhibit torsion during movement, rotating from a partially concave backward orientation, thru concave down to partially concave forward during protraction (*Knutsen et al., 2008*). We thus define positive and negative $\theta_p$ to denote whisker movement in the protraction and retraction directions respectively, independent of intrinsic curvature.

Slip-offs were more likely for more distant object locations (*Figure 5B*) and occurred at larger $\theta_p$ for smaller object distances (*Figure 5C*). Overall, slip-offs occurred in approximately 15% of behavioral trials. We again compared model and experiment in the $\theta_p-d$ plane. One of the two whiskers was truncated (*Figure 5C*, left). For objects touching the whisker near the tip, detachments occurred for small $\theta_p$ (<20°), with the whisker tip reaching the pole (i.e., pull-offs). For smaller object distances slip-off occurred at larger $\theta_p$ (>20°), consistent with a saddle-node bifurcation (i.e., slip-offs). The second whisker was less truncated (*Figure 5C*, right). For the object distances tested we observed slip-offs only along the whisker. These results are consistent with our model. The critical $\theta_p$ values for slip-off varied significantly across trials even for identical object distances. This variability is likely caused by differences in whisker movement and whisker elevation across trials.

Rodents move their whiskers over objects to explore surfaces. For example, mice can discriminate surface roughness over a few whisking cycles (*Chen et al., 2013*). Texture is likely inferred from the statistics of whisker micromotions produced by the interactions between whiskers and objects (*Diamond, 2010*). In particular, as whiskers move over objects whiskers occasionally get stuck, followed by

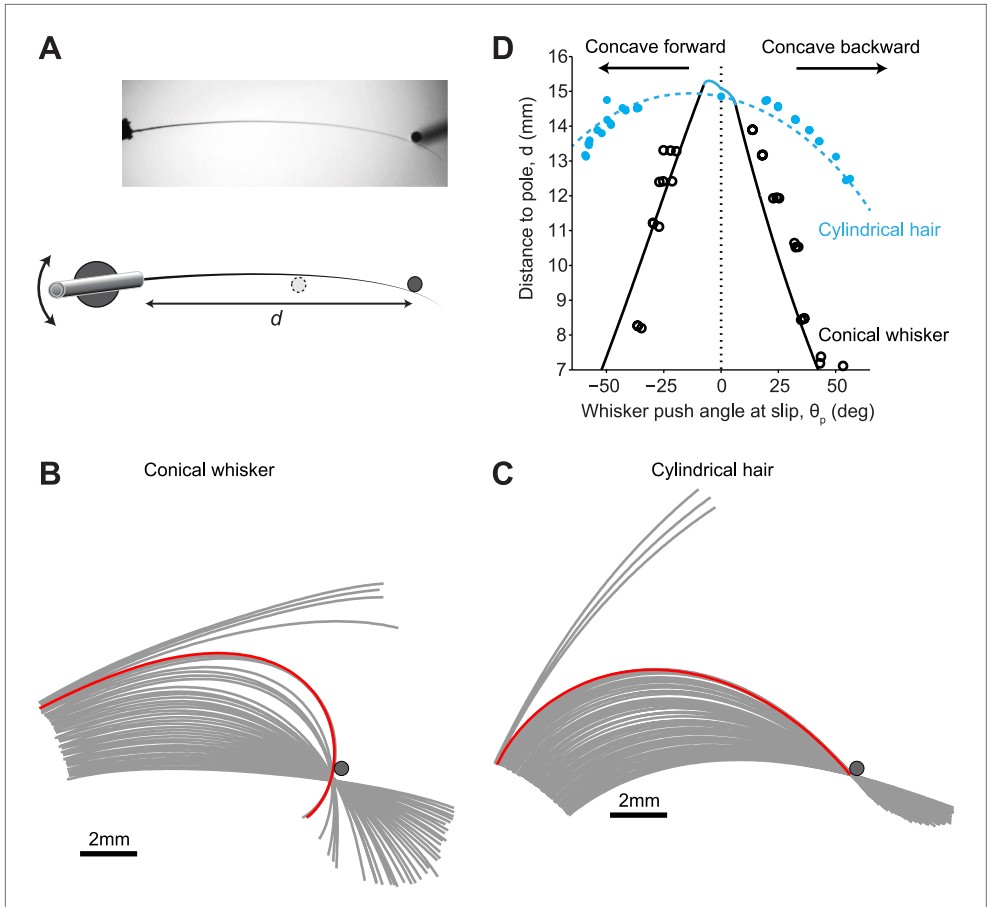

**Figure 4**. Isolated whiskers interacting with cylindrical poles. (**A**) Top-down view of a mouse C2 whisker mounted on a galvanometer scanner. The scanner rotates the whisker into a vertical pole. The distance of the pole from the base of the whisker, $d$, is varied across experiments. (**B**) Snapshots of the whisker at 32 Hz as it is smoothly rotated (0.2 Hz, counter clockwise) into and past the pole. Red line, whisker shape immediately (<32 ms) before slip-off. Note that the end of the whisker had not reached the point of object contact. (**C**) Snapshots of a near-cylindrical hair. Red line, hair shape immediately before pull-off. Note that the end of the hair had reached the point of object contact. (**D**) The detachment curve in the $\theta_p - d$ plane. Solid line, theoretical prediction for conical whisker; open circles, experimental measurements for conical whisker. Dashed line, theoretical prediction for cylindrical hair; solid circles, experimental measurements for cylindrical hair. Blue, pull-offs occur because whisker tip has reached the object. Black, slip-offs occurs because of saddle-node bifurcation. Parameters of the conical whisker: $L_w$ = 15.25 mm, $r_{base}$= 32.5 μm, $r_{tip}$ = 2 μm, A = 0.02 mm$^{-1}$. Parameters of the approximately cylindrical hair: $L_w$ = 15.0 mm, $r_{base}$= 30 μm, $r_{tip}$ = 26.5 μm, A = 0.017 mm$^{-1}$. Pole radius, $r_{pole}$ = 0.25 mm.

high-velocity slips. The pattern of stick-slip events is highly informative about surface texture (**Arabzadeh et al., 2005**; **Wolfe et al., 2008**).

We wondered whether whisker shape determines the nature of the stick-slip events underlying texture exploration. We moved a C2 mouse whisker over extra fine (600 grit) sandpaper using a galvanometer scanner ($f_{galvo}$ = 0.2 Hz; peak-to-peak amplitude, 30°) while tracking whisker shape in three dimensions using dual view videography (**Figure 6A**, **Video 3**). The tips of mouse whiskers moved along the surface in an irregular manner, during protractions and retractions. Whisker tips were transiently trapped (**Figure 6B**, red) followed by small, high-velocity slips (**Figure 6B,C**). The pattern of stick-slip events differed for different wall distances (**Figure 6C**).

In contrast, as cylindrical hairs swept across the surface they were trapped during initial protraction and remained trapped for the remainder of the experiment lasting multiple whisking cycles (**Figure 6D–F**, **Video 4**). When the distance between follicle and the site of trapping was

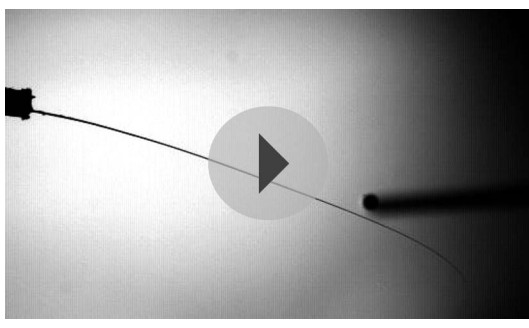

**Video 1**. Example video of a conical whisker mounted on the galvo (*Figure 4*) slipping off a pole. Speed 16fps, 0.5x real-time.

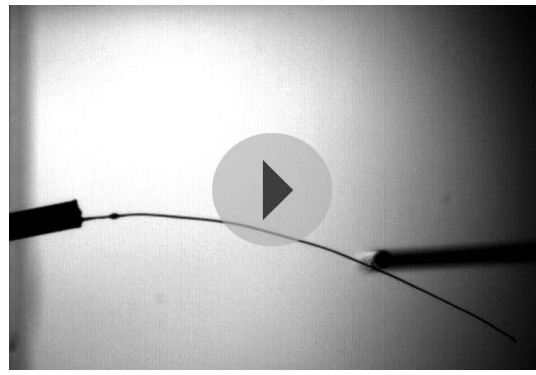

**Video 2**. Example video of a cylindrical hair mounted on the galvo (*Figure 4*) pulling off a pole. Speed 16fps, 0.5x real-time.

shorter than the hair length, the hair buckled out of the plane of movement (*Figure 6E*, top). The tips of hairs escaped the traps only when the distance to the tip along the path of an undeflected hair exceeded the actual hair arclength (*Figure 6E,F*, pane 2). The whisker tip was thus pulled out of the trap (*Figure 3J*, blue line). These measurements show that the conical whisker shape is critical for the sweeping motions of whisker tips across objects and surfaces, which supports feature extraction via stick-slip events. More generally, conical whiskers can move past walls and objects, which may be critical for positioning of whiskers in confined spaces, such as tunnels, during directed tactile exploration.

We investigated whether slip-offs convey specific sensory information to cortex. Silicon probes were inserted into the C2 barrel column (*O'Connor et al., 2013*) (*Figure 7A*). We recorded multi-unit activity across cortical layers 2–5 while mice performed an object location discrimination task with the C2 whisker (*O'Connor et al., 2013*). Mice touched the pole multiple times during a trial (*Figure 7B*). The first touch within a trial caused a large peak in activity with a rapid rise (*Figure 7B,C*), consistent with previous work (*Simons, 1978*; *Armstrong-James et al., 1992*; *de Kock et al., 2007*; *O'Connor et al., 2010b*; *O'Connor et al., 2013*). Later touches within a series, during which slip-offs were more commonly seen, produced smaller responses (*Figure 7D*) (*Ahissar et al., 2001*). When slip-off did not occur, the detach-related signals were almost undetectable. In contrast, when slip-off did occur, the detach-related signals were large, comparable to the first touch (*Figure 7F,G*).

## Discussion

We have developed a mathematical framework for whisker deflection in the context of dynamical systems theory (*Figures 1–3*) to explore the functional consequences of whisker taper. Recent findings have shown that whisker taper is used as a ruler by mice to gauge the distance to objects with a single whisker (*Pammer et al., 2013*). Tapered whiskers have resonance frequencies that are robust to wear and truncation damage to their tips (*Williams and Kramer, 2010*). Tapered whiskers also detach from objects at shallower push angles than cylindrical whisker substitutes.

Here we go beyond prior observations and uncover fundamental differences in how tapered and untapered hairs interact with objects; tapered whiskers slip-off when the contact point is along the whisker body ($s_{obj} < L_w$), whereas cylindrical hairs require the tip to be pulled off ($s_{obj} = L_w$) for biologically plausible push angles (*Figure 3E,J*, *Figure 4*). Thus, tapered whiskers have greater freedom of movement past obstacles compared to cylindrical hairs. This mobility is seen in reduced preparations (*Figure 4*) and also during active behavior, as whisker slip-off occurs in a variety of object location discrimination tasks (*Figures 5, 7*). The intrinsic difference in detachment also produces qualitatively different interaction patterns during palpation of textured surfaces (*Figure 6*). Tapered whiskers sweep past with stick-slip micromotions, whereas cylindrical whiskers become immobilized on surface imperfections.

Theoretical treatment of whisker mechanics is a necessary foundation for understanding how sensory input shapes neural representations of the tactile world. Previous work has computed

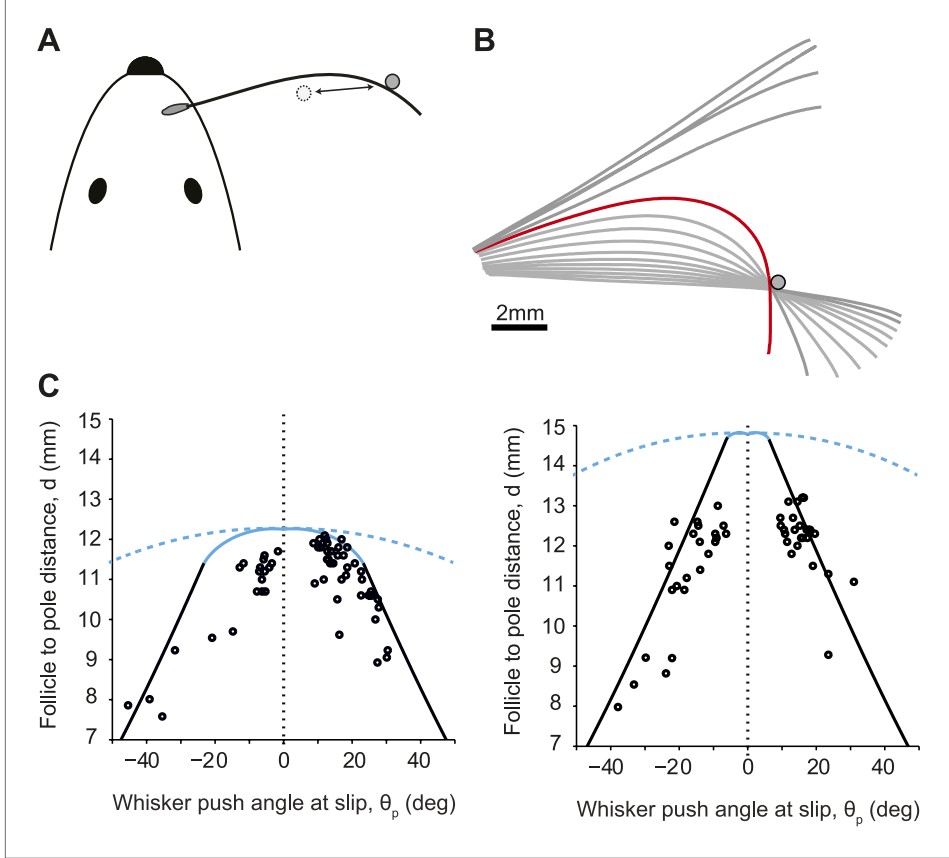

**Figure 5**. Slip-offs during object location discrimination behavior. (**A**) Schematic of a mouse whisking to touch a pole (experiments from **Pammer et al., 2013**). (**B**) Time series (250 Hz) of whisker shape around example protraction slip event. Frame of slip-off is highlighted in red. (**C**) Detachment curves in the $\theta_p - d$ plane for two whiskers. Solid line, theoretical predictions for conical whisker; open circles, experimental measurements for conical whiskers. Dashed line, theoretical predictions for cylindrical hair. Blue, pull-offs. Black, slip-offs. Left, truncated whisker with parameters: $L_w$ = 12.5 mm, $r_{base}$ = 35 μm, $r_{tip}$ = 8.5 μm. Right, whisker parameters: $L_w$ = 15.3 mm, $r_{base}$ = 33.5 μm, $r_{tip}$ = 2 μm. For both whiskers, intrinsic curvature was $y$ = A(x−2.2 mm)$^2$ where A = 0.02 mm$^{-1}$.

whisker deflections based on the quasi-static solution of the Euler-Bernoulli equation (**Birdwell et al., 2007**; **Williams and Kramer, 2010**; **Quist and Hartmann, 2012**; **Pammer et al., 2013**). Aspects of whisker vibrations have also been treated, including resonant frequencies (**Hartmann et al., 2003**; **Neimark et al., 2003**) and wave propagation following contact-induced impulses (**Boubenec et al., 2012**).

We framed whisker-object interactions in the language of boundary-value problems. This allowed us to carry out bifurcation analysis and distinguish stable from unstable shapes. We demonstrate that for conical whiskers there are only two possible solutions for whisker shape for a given object distance and push angle, one stable, one unstable. We identify how the saddle-node bifurcations separating the two branches of solutions vary as a function of parameters, such as $\theta_p$ (**Figure 3**). This is not possible using previously developed numerical approaches (**Birdwell et al., 2007**; **Quist and Hartmann, 2012**). Slip-offs occur suddenly at a critical push angle, $\theta_p$, corresponding to the angle where these saddle-node bifurcations occur. Curves of saddle-node bifurcations in a two-parameter plane define the regime in which stable solutions can be obtained (**Figure 3E,J**). For conical whiskers the critical push angles are within the normal range of whisking (**Figure 4**). For cylindrical hairs the critical angles fall outside the range of whisking. Thus our theory predicts that conical and cylindrical whiskers will interact with objects in a fundamentally different manner.

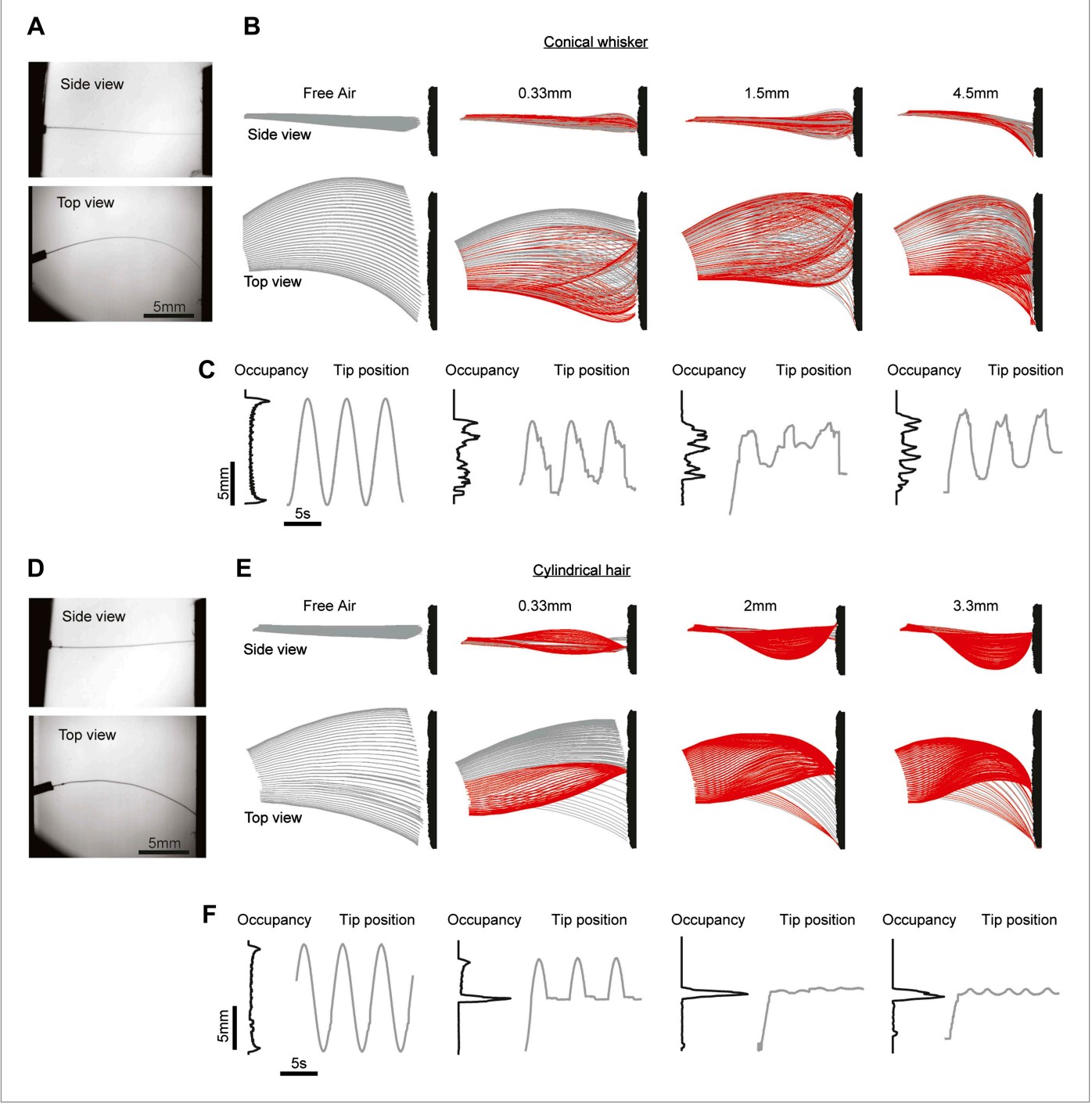

**Figure 6**. The whisker taper is necessary for slips across textures. (**A**) Dual-perspective imaging of a conical whisker, mounted on a galvo, sweeping across a texture (600 grit sandpaper). 'Top', Side view; 'Bottom', Top view. (**B**) Conical whisker swept past the texture at four distances: Free air, push distance $d_p$ = 0.33 mm, 1.5 mm and 4.5 mm. $d_p = \|x(L_w)-x(0), y(L_w)-y(0)\|-d$, where $d$ is the nearest distance from the base of the whisker to the surface. In other words, $d_p$ is the distance the surface is moved radially into the whisker beyond just touching. Red traces indicate frames where the whisker tip is stuck, gray traces where the tip is slipping along the surface. Surface texture is schematic and exaggerated. (**C**) Black lines, histograms of tip position over time. Gray lines, trajectories of the whisker tip over the first three whisking periods. Traces are aligned to peak of theta at base. (**D–F**), as (**A–C**), but using a cylindrical hair of similar length. Free air, push distance $d_p$ = 0.33 mm, 2 mm and 3.3 mm. Whisker parameters $L_w$ = 16.4 mm, $r_{base}$ = 33.5 µm, $r_{tip}$ = 2 µm. Hair parameters as in **Figure 4**.

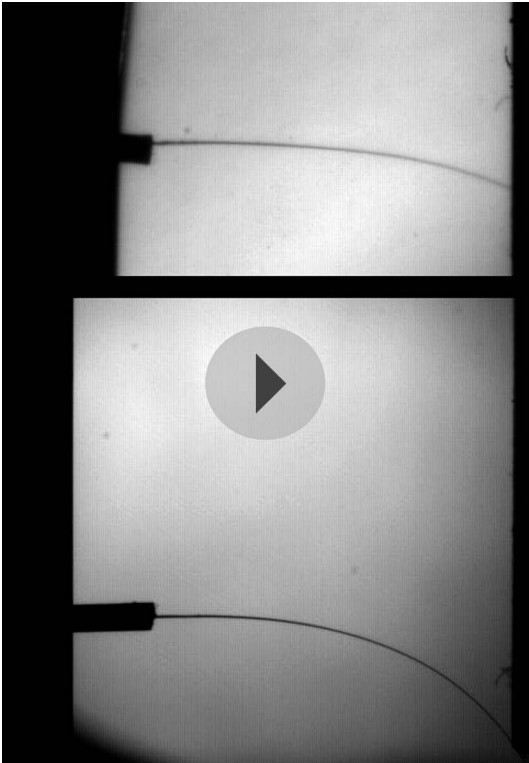

**Video 3**. Composite video of the conical whisker mounted on a galvo slipping across the textured surface in **Figure 6B** (4.5 mm push distance). Upper video is the side view, lower video is the top view. Speed 32fps, 1x real-time.

Our theory addresses the effects of whisker truncations (**Figures 3–5**). Truncations of conical whiskers make whisker behavior more 'cylindrical-like' (**Figure 5**). The intuition obtained from our analysis led us to distinguish between the dynamics of conical whiskers and cylindrical hairs during sweeping across textures (**Figure 6**).

We compared theoretical predictions with videos of whiskers and cylindrical hair rotated into a steel pole. In situations where the whisker curvature was contained within a plane, the agreement between theory and experiment was very good (**Figure 4**), despite our model ignoring frictional forces. The small remnant differences between theory and experiment are due to deviations of whisker geometry from perfect conical shape (AH, KS, DG, unpublished) and possible inhomogeneities in the Young's modulus (**Quist et al., 2011**) (but also see **Carl et al., 2012**). In behaving mice the whisker droops out of the plane of whisking (**Towal et al., 2011**). Rigorous treatment of whisker deflection by an object would thus require a three-dimensional model. We developed a phenomenological model to predict slip-offs even for behaving mice ('Materials and methods'), which produced qualitative agreement with experiments (**Figure 5**).

Our results suggest several functions for which the conical shape of rodent whiskers is evolutionarily adaptive. Within their natural habitat, many rodents, including house mice (**Berry, 1968**) and African pouched mice (**Ellison, 1993**), live in group nests consisting of chambers connected by long, body-width tunnels. During running, whiskers point forward to touch unanticipated objects. When a new object is encountered rodents foveate their whiskers on the object for fine-scale exploration (**Grant et al., 2009**). Within these dark, radially constrained tunnels, navigation (**Vincent, 1912**; **Dehnhardt et al., 2001**), localization of objects (**Hutson and Masterton, 1986**; **Krupa et al., 2001**), social touch (**Wolfe et al., 2011**), and determination of friend or foe (**Anjum et al., 2006**) demands freedom of whisker motion. Whiskers have to be moved past the rough walls of the tunnel. Without the flexibility provided by whisker taper, the whiskers could be trapped in a far protracted or retracted orientation, causing a tactile 'blind-spot'.

Whisker taper is also desirable outside of constrained spaces. A major sensory avenue for the localization and identification of objects and their properties is via directed sweeping of whiskers across object surfaces (**Carvell and Simons, 1990**; **Ritt et al., 2008**; **von Heimendahl et al., 2007**; **Wolfe et al., 2008**). During artificial periodic palpation of fine-grained textures, conical whiskers traversed the surface with complex micromotions, whereas cylindrical hair became trapped against the surface (**Figure 6**). Although a precise understanding of the interaction between a tapered whisker and textured surface during a stick-slip event has not been treated mathematically, it is likely that forces at the tip build up until they bend the whisker tip sufficiently to free it from traps. In the cylindrical case, the constant bending stiffness of the body and tip render the whisker incapable of transmitting sufficient lateral force to buckle the much stiffer tip and release it from the surface.

Beyond mechanical maneuverability, do slips contribute to the neural representation of tactile sensation? During active whisking, stick-slip micromotions on textured surfaces drive sparse, precisely timed spikes in barrel cortex that provide a sensory cue for surface texture (**Jadhav et al., 2009**). Neural responses in barrel cortex to repeated contacts between whiskers and objects show strong adaptation during active touch (**Ahissar et al., 2001**; **Crochet et al., 2011**) and object location

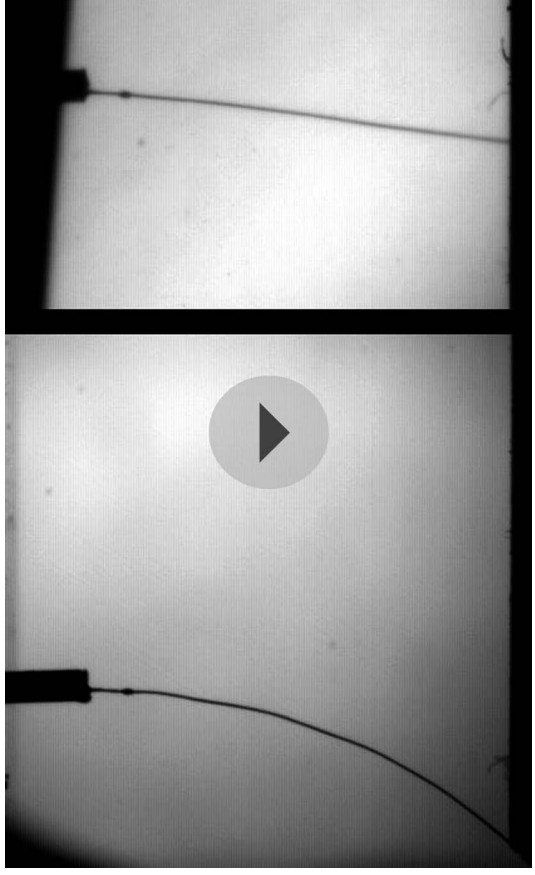

**Video 4**. Composite video of the cylindrical hair mounted on a galvo getting stuck on the textured surface in *Figure 6E* (3.3 mm push distance). Upper video is the side view, lower video is the top view. Speed 32fps, 1x real-time.

discrimination (*Figure 7C,D,G*). Despite occurring when the circuit is adapted to touch, slip-offs produce strong volleys of cortical activity, of comparable magnitude to pre-adapted touch (*Figure 7F*). Thus, slip-related excitation can overcome cortical touch adaptation and likely contributes to sensation and perception in a variety of tactile behaviors (*Arabzadeh et al., 2005*).

## Materials and methods

### Model of a whisker deflected by a cylindrical pole

We model whiskers as truncated cones with length $L_W$, base radius $r_{base}$, and tip radius $r_{tip}$ (*Figure 1A*). The conical shape is virtually extended to a perfect cone of length $L$. The whisker is located in the *x-y* plane. The arclength along the whisker, *s*, is $s = 0$ at the base, $s = s_{obj}$ at the point of object contact, $s = L_W$ at the tip, and $s = L$ at the virtual tip (*Figure 1B*). The whisker base is located at point $(x_0, y_0)$, and the positions of a point along the whisker is $(x(s), y(s))$, $0 \le s \le L_W$. The running angle between the whisker and the *x*-axis is $\theta(s)$, and $\theta(0) = \theta_0$. The whisker radius is $r_w = (L-s) r_{base}/L$ and the area moment of inertia is $I(s) = \dfrac{\pi r_w^4}{4}$. The Young's modulus is $E = 3$ GPa (*Birdwell et al., 2007*; *Quist et al., 2011*; *Pammer et al., 2013*). Similar calculations were carried out for cylindrical hair with $r_w = r_{base}$.

The bending stiffness of the whisker is the product $EI(s)$. In the absence of contact with an object, the intrinsic curvature of the whisker is $\kappa_i(s)$. The object is a cylindrical pole oriented perpendicular to the *x-y* plane with radius $r_{pole}$, centered at $(x_{cen}, y_{cen})$. Upon contact, the whisker touches the object at $(x_{obj}, y_{obj})$ with angle $\theta_{obj}$ (*Figure 1B*), where

$$x_{obj} = x_{cen} - r_{pole} \sin\theta_{obj} \qquad y_{obj} = y_{cen} + r_{pole} \cos\theta_{obj} \qquad (1)$$

The Euclidian distance between the whisker base and the contact point is *d*. The object applies force $\vec{F}$ on the whisker:

$$\vec{F} = (F_x, F_y) = \left( -F \sin\theta_{obj}, F \cos\theta_{obj} \right) \qquad (2)$$

At steady state, the shape of the whisker is determined by the solution of the static Euler-Bernoulli equation (*Landau and Lifshitz, 1986*; *Birdwell et al., 2007*; *Williams and Kramer, 2010*; *Pammer et al., 2013*)

$$\frac{d\theta}{ds} = \kappa_i(s) + \frac{M_z(s)}{EI(s)} \qquad (3)$$

where $M_z$ is the component of the bending moment $\vec{M} = \vec{r} \times \vec{F}$ perpendicular to the *x-y* plane and $\vec{r}(s) = \left( x_{obj} - x(s), y_{obj} - y(s) \right)$, together with the equations

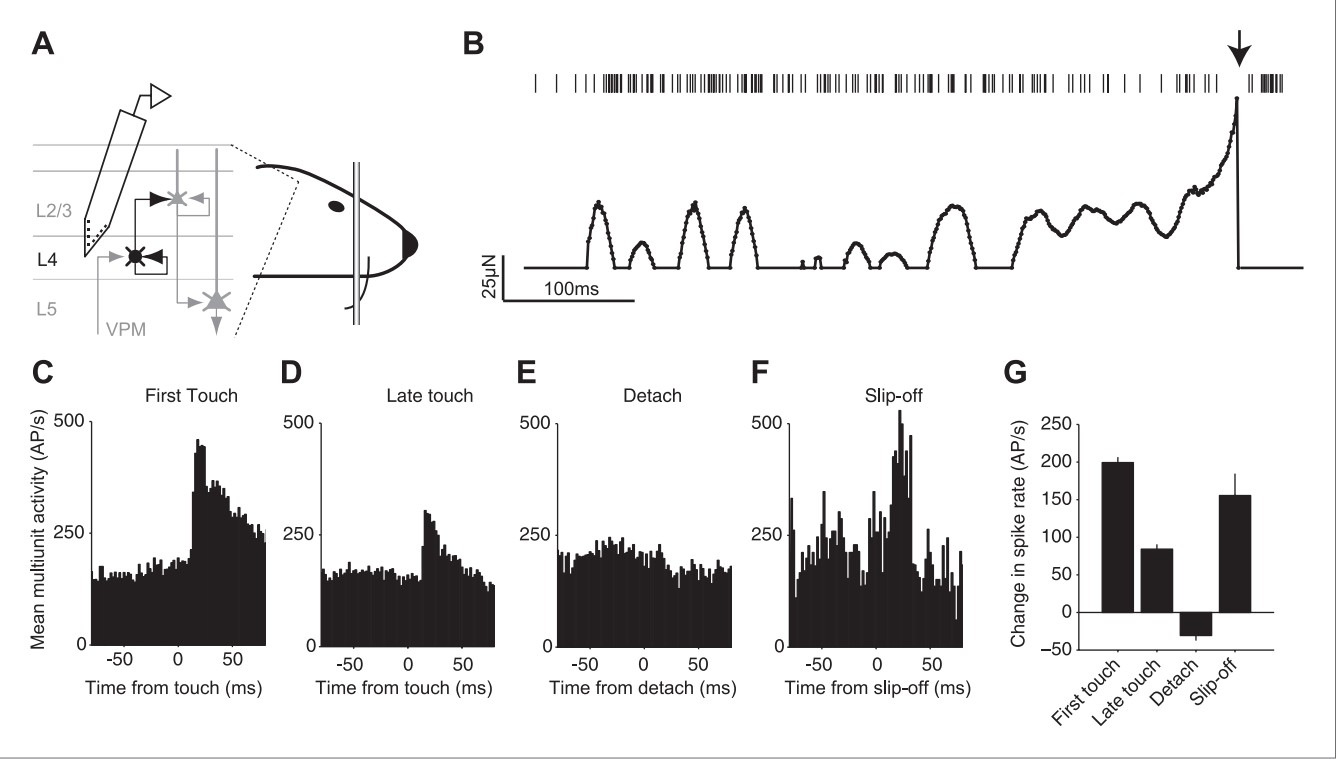

**Figure 7**. Neural signals of slip-off in the barrel cortex. (**A**) Silicon probe recording during a pole localization task (experiments from ***O'Connor et al., 2013***). (**B**) Spikes and whisker forces, one behavioral trial. 'Top', multi-unit activity. Arrow, slip-off event. 'Bottom', contact induced forces. Solid circles, time points with non-zero contact-mediated forces calculated from changes in whisker curvature. A 2 ms period during which the whisker was slipping off was removed as the quasi-static model is invalid for such highly dynamic events. (**C**–**F**) Multi-unit spike responses in the barrel cortex (shank < 300 μm from the center of the C2 barrel). (**C**) Activity aligned to the first touch in a trial (720 responses; two animals, three sessions, six electrode shanks). (**D**) Same as (**C**), but aligned to the last touch before a behavioral response (i.e., lick) (720 responses). (**E**) Same as (**C**), but aligned to the moment of detach on the last touch before a behavioral response in trials without slip-offs (392 responses). To prevent contamination by touch-onset this analysis was restricted to touches that were longer than 50 ms. (**F**) Aligned to slip-off (34 responses). (**G**) Change in spike rate triggered by the event (activity 10–30 ms post event, minus activity −50 to 0 ms pre event). Error bars, SEM. Pairwise comparison showed no significant difference in evoked spikes between first touch and slip-off groups (p=0.20), a significant difference between last touch and slip-off (p=0.037) and significant differences between all other groups (every pair, p<10⁻¹²).

$$\frac{dx}{ds} = \cos\theta \tag{4}$$

$$\frac{dy}{ds} = \sin\theta \tag{5}$$

Substituting ***Equations 1,2*** in ***Equation 3***, we obtain

$$\frac{d\theta}{ds} = \kappa_i(s) + \frac{F}{EI(s)}\left[\left(x_{cen} - r_{pole}\sin\theta_{obj} - x\right)\cos\theta_{obj} + \left(y_{cen} + r_{pole}\cos\theta_{obj} - y\right)\sin\theta_{obj}\right] \tag{6}$$

We seek a solution for ***Equations 4–6*** given the boundary conditions at the base ($x_0$, $y_0$ and $\theta_0$), and that the whisker contacts the pole at an (initially unknown) arclength $s_{obj}$.

Given the shape of an undeflected whisker as a function of the running arclength $s$, namely $(x,y)=(g(s),h(s))$, the intrinsic curvature is

$$\kappa_i = \frac{g'(s)h''(s) - h'(s)g''(s)}{\left\{[g'(s)]^2 + [h'(s)]^2\right\}} \tag{7}$$

where $d/ds$ is denoted by '. The shape of the undeflected whisker is considered to be parabolic, $y = Ax^2$ (***Quist and Hartmann, 2012***).

To compute the whisker shape ***Equations 4–7*** are transformed to a form of a boundary-value problem (BVP) by introducing a variable $\sigma = s/s_{obj}$

$$\frac{dx}{d\sigma} = s_{obj}\cos\theta \tag{8}$$

$$\frac{dy}{d\sigma} = s_{obj}\sin\theta \tag{9}$$

$$\frac{d\theta}{d\sigma} = s_{obj}\frac{g'(s_{obj}\sigma)h''(s_{obj}\sigma) - h'(s_{obj}\sigma)g''(s_{obj}\sigma)}{\left\{\left[g'(s_{obj}\sigma)\right]^2 + \left[h'(s_{obj}\sigma)\right]^2\right\}^{\frac{3}{2}}}$$
$$+ \frac{s_{obj}F}{E(s_{obj}\sigma)I(s_{obj}\sigma)}\left[(x_{cen} - r_{pole}\sin\theta_{obj} - x)\cos\theta_{obj} + (y_{cen} + r_{pole}\cos\theta_{obj} - y)\sin\theta_{obj}\right] \tag{10}$$

$$\frac{dF}{d\sigma} = 0 \tag{11}$$

$$\frac{d\theta_{obj}}{d\sigma} = 0 \tag{12}$$

$$\frac{ds_{obj}}{d\sigma} = 0 \tag{13}$$

The differential ***Equations 8–13*** are solved on the interval $0 \leq \sigma \leq 1$ together with the equations

$$I(s_{obj}\sigma) = \frac{\pi}{4}\left(\frac{L - s_{obj}\sigma}{L}\right)^4 \tag{14}$$

The boundary-value conditions for $\sigma = 0$ are: $x(0) = x_0$, $y(0) = y_0$, $\theta(0) = \theta_0$. The conditions for $\sigma = 1$ are: $x(1) = x_{cen} - r_{pole}\sin\theta_{obj}$, $y(1) = y_{cen} + r_{pole}\cos\theta_{obj}$, $\theta(1) = \theta_{obj}$. Solutions to ***Equations 8–14*** have physical meaning if $s_{obj} < L_W$. If the whisker tip reaches the contact point ($s_{obj} = L_W$) the whisker detaches because it is pulled off the pole.

We solved six differential ***Equations 8–13*** together with their boundary conditions to find six unknown variables ($x$, $y$, $\theta$, $F$, $\theta_{obj}$, $s_{obj}$) as functions $\sigma$ on the interval $0 \leq \sigma \leq 1$. The variable $\theta_{obj}$ is treated as a separate variable from $\theta$, but the boundary condition $\theta(1) = \theta_{obj}$ guarantees that the solution is self-consistent. The equations were solved numerically using the iterative shooting method (***Press et al., 1992***). We start with guessed initial values for the unknown variables for $\sigma = 0$ and integrate the differential equation until $\sigma = 1$. The initial conditions are then varied to reduce the difference between the given boundary conditions and those that are obtained by the most recent integration. The method converges if the initial conditions are sufficiently close to the solution. We begin by solving the boundary-value problem with $\theta_{obj}$ corresponding to $\theta_p = 0$ (i.e., the whisker is barely touching the pole). We then vary $\theta_{obj}$ slightly, compute the whisker shape, and repeat the process until the desired $\theta_{obj}$ is reached. We used the boundary-value problem solver software package XPPAUT (***Ermentrout, 2002***). The software package AUTO (***Doedel, 1981***), which is incorporated into XPPAUT, was used to compute bifurcation diagrams (***Figure 3***), by following the solutions of the boundary-value problem as parameters, such as $\theta_{obj}$, vary.

The static solution of ***Equations 8–14*** is a fixed point of a spatiotemporal dynamical system representing whisker movement. The full dynamical system can be formulated as a partial differential equation only for small $\theta_p$ and straight beams (***Timoshenko, 1961***; ***Boubenec et al., 2012***). Since the full dynamical system for all $\theta_p$ and beams with intrinsic curvature is not known we cannot linearize a dynamic equation. However, the static solution for small $\theta_p$ must be stable. In addition, bifurcation theory

implies that if we increase $\theta_p$ the solution will coalesce with an unstable solution and they both disappear, via a saddle-node bifurcation (*Strogatz, 1994*). In principle, a branch of stable solutions can lose stability via a Hopf bifurcation before the saddle-node bifurcation. Slip-off will occur at $\theta_p$ values smaller than predicted by our quasi-static theory. The good correspondence between the computed saddle-node bifurcation and the experimentally measured value for $\theta_p$ at slip-off shows that the static solution disappears via a saddle-node bifurcation (*Figure 4*).

Undeflected whiskers can be modeled as parabolas within a plane (*Towal et al., 2011*). In the work reported here the whisker is contained entirely within a plane perpendicular to the pole. For a whisker with intrinsic curvature, contact occurs in either the 'concave backward' (CB) or 'concave forward' (CF) directions (*Figure 1C*) (*Quist and Hartmann, 2012*). To quantify contact strength, we use the push angle $\theta_p$ (*Figure 1D*) (*Quist and Hartmann, 2012*). Suppose a whisker originates at $(x_0, y_0, \theta_0)$ and touches a pole at $(x_{obj}, y_{obj})$. We plot an undeflected whisker with the same $(x_0, y_0, \theta_0)$, and find a point along the whisker with the same Euclidian distance $d$ from $(x_0, y_0)$ as $(x_{obj}, y_{obj})$, defined as $(x_{virtual}, y_{virtual})$ The angle between the two rays starting at $(x_0, y_0)$ towards $(x_{obj}, y_{obj})$ and $(x_{virtual}, y_{virtual})$ is defined as $\theta_p$. By convention, we define the sign of $\theta_p$ to be positive for CB and negative for CF.

Exact treatment of whisker deflection in behaving rodents demands a three-dimensional model that is outside the scope of this work. Instead, we developed a phenomenological two-dimensional model (*Figure 5*). First we assume that the whisker touches the object in a concave-down configuration. Second, the deflection of the whisker is described by the two-dimensional model (*Equation 8–14*) when the projection of the whisker on that plane is treated as a two-dimensional whisker. The area moment of inertia ($I$) is computed by estimating the arclength $s$ along the real whisker from the whisker projection and using this value in *Equation 14*. This correction in $s$ was on the order of 3%. We measured the length of the isolated whisker. Estimating the whisker base is inaccurate because of the fur on the face. We therefore determined the effective whisker length from the estimated whisker base to the tip based on the video recordings. If the whisker slips off at its tip, we find the maximal projected length during events of slip-off at the tip. If there are no such events, we compute the projected length that yields the theoretically-obtained slip-off at the largest $d$ for which slip-off is obtained. For all cases, this estimated value is less than 1 mm smaller than the length measured directly.

## Whisker measurements

For galvo experiments (*Figures 4, 6*), we used plucked, full-grown mouse C2 whiskers. The shapes of these whiskers were measured under a light microscope at high magnification (*Pammer et al., 2013*). The follicle ends of the whiskers were embedded in the barrel of a cut 21 gauge needle filled with Kwik-Cast silicon sealant (World Precision Instruments, Sarasota, FL). Needles were mounted on the top edge of a galvo scan mirror (6800HP; Cambridge Technology, Bedford, MA). Whiskers were then rotated into a cylindrical object (steel Wiretrol II plunger; Drummond Scientific, Broomall, PA) at 0.2 Hz, 30° peak-to-peak amplitude (*Figure 4*). Dual-perspective imaging confirmed that whiskers remained in the concave forward or concave backward orientation during the interaction with the pole (data not shown). The same whiskers were used for imaging whisker motion across textured surfaces (*Figure 6*). The surface was fine sandpaper (600 grit) rigidly mounted on a glass slide and positioned perpendicular to both planes of imaging. A variety of human hair was characterized. Hair from an Asian female closely matched the whisker diameter close to the base and was used as a cylindrical hair. The hair dimensions were: base diameter, 60 μm; tip diameter, 53 μm; length, 15.0 mm.

High-speed videography was used to measure the position and shape of mouse whiskers during galvo experiments (*Figures 4, 6*) (X-PRI camera, 32 fps, 0.6 ms exposure, 8-bit depth, AOS Technologies, Switzerland) and behavior (*Figures 5, 7*) (1000 fps, 0.2 ms exposure, 8-bit depth, Basler 504 k, Germany). Pixel size was 0.07 mm (*Figure 5*), or 0.031 mm (*Figure 7*), or 0.032 mm (*Figures 4, 6*). Illumination was with a 940 nm infrared LED delivered through a diffuser and condenser lens and projected directly into the camera. A silver mirror (PFSQ10-03-P01, Thor Labs, Newton, NJ) reflected an orthogonal side view projection onto the same camera (*Knutsen et al., 2008*). Videos were split and cropped prior to whisker tracking.

Whiskers were tracked with the Janelia Whisker Tracker (*Clack et al., 2012*) (https://openwiki.janelia.org/wiki/display/MyersLab/Whisker+Tracking). The whisker medial axis is stored as an array of points $(x_i, y_i)$, i = 1,…,N, where $N$ is on the order of several hundreds. To remove artifacts associated with tracking variation at the base when calculating $\theta_0$, the angle of the whisker base was determined by a linear fit of the fifth through tenth points closest to the base. Forces acting on the whisker (*Figure 7B*) were calculated using published methods (*O'Connor et al., 2010a*; *Clack et al., 2012*; *Pammer et al., 2013*).

The behavioral task and apparatus have been described in detail elsewhere (*O'Connor et al., 2010a*; *O'Connor et al., 2013*; *Pammer et al., 2013*). Briefly, head-fixed mice judged the distance to a metal pole that was presented at a range of positions along the whisker in the radial dimension (*Figure 5*) or horizontal dimension (*Figure 7*). For radial discrimination, a proximal position (5 mm radially from follicle) was defined as the Go position, distal positions (7–13 mm) were defined as No Go positions. For horizontal discrimination, Go and No go positions were separated by 4.5 mm along a parallel to the anteroposterior axis of the mouse at a radial distance of 7–11 mm. Mice performed object location discrimination with a single C2 whisker. Within two days of the behavioral experiments we plucked whiskers and measured their shape and material properties using a macroscope and microgram balance (Mx5; Mettler Toledo, Columbus, OH) (*Pammer et al., 2013*).

## Electrophysiology

Parts of the electrophysiology dataset is a reanalysis of previously acquired data (see *O'Connor et al., 2013* for detailed methods). During a head-fixed pole location discrimination task, a 32 channel, four shank silicon probe (Buz32, Neuronexus, Ann Arbor, MI) was lowered into barrel cortex, with an estimated tip depth of 375–720 µm from the pia. Prior to insertion, probes were painted with DiI. Following recordings, mouse brains were fixed, stained for cytochrome oxidase and tangentially sectioned to determine the location of the shanks within the barrel field. Shanks within 300 µm of the center of C2 were included for analysis for slip-off responses (two animals, three behavioral sessions, six shanks). Following common signal subtraction, bandpass filtering between 300 and 6,000 Hz, spike extraction of 4 s.d. threshold crossing, and spike merging, multiunit responses were aligned to whisker behavioral events. Spikes with peaks <307.5 µs jitter on the same shank were considered a single spike. Each multiunit was the sum of activity on all eight electrodes on a single shank (six total multiunit recordings). Slip-off events were rare (17 total in three sessions) compared to detach without slip-off. Significance was calculated as unpaired two-tailed t-tests on the difference between the number of spikes in the period 10–30 ms post event and the 50 ms prior to event normalized to the respective period lengths followed by Bonferroni–Holm correction for multiple comparisons.

## Acknowledgements

The authors thank Mitra Hartmann, Brian Quist, Lucie Huet and Tansu Celikel for useful discussions, Diego Gutnisky, Nick Sofroniew and Daniel O'Connor for comments on our manuscript.

## Additional information

### Funding

| Funder | Grant reference number | Author |
|---|---|---|
| Howard Hughes Medical Institute | | Samuel Andrew Hires, Lorenz Pammer, Karel Svoboda, David Golomb |
| Israel Science Foundation | 88/13 | David Golomb |

The funders had no role in study design, data collection and interpretation, or the decision to submit the work for publication.

### Author contributions

SAH, DG, Conception and design, Acquisition of data, Analysis and interpretation of data, Drafting or revising the article; LP, Acquisition of data, Analysis and interpretation of data; KS, Conception and design, Analysis and interpretation of data, Drafting or revising the article

### Ethics

Animal experimentation: This study was performed in strict accordance with the recommendations in the Guide for the Care and Use of Laboratory Animals of the National Institutes of Health. All procedures were in accordance with protocol 08–42 approved by the Janelia Farm Research Campus Institutional Animal Care and Use Committee. All surgical procedures were performed under isoflurane

anesthesia, and every effort was made to minimize suffering including administration of buprenorphine, ketoprofen and marcaine during and after surgery.

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
