## [Decision Letter]

Thank you for sending your work entitled “Tapered whiskers are required for active tactile sensation” for consideration at *eLife*. Your article has been favorably evaluated by a Senior editor and 2 reviewers, one of whom is a member of our Board of Reviewing Editors. As described below, the requested revisions are quite minor and we should be able to make a final decision in a very short time after receipt of your revision.

The Reviewing editor and the other reviewer discussed their comments before we reached this decision, and the Reviewing editor has assembled the following comments to help you prepare a revised submission.

It is well established that rodent and cat whiskers are conical in contrast to human hairs and seal whiskers. A fundamental question, unanswered for many years, is whether the conical shape of the formers has a functional significance.

This paper provides an important contribution in the answer to that question. This work is the most comprehensive study of the relationship between the shape of a whisker and the profile it takes upon touch to a small object or a texture. Combining analytical, numerical, and experimental approaches, the authors demonstrate that the profile depends crucially on the whisker's tapering. A central result here is that when a conically tapered whisker moves and contact an object, it can slip off the object well before the latter has touched the tip of the whisker. To get some insight on the possible functional relevance of this effect, the authors complete their study by performing electrophysiological recordings in S1 of behaving mice while monitoring the whiskers' motion. They confirm previous studies, which showed that S1 neurons respond strongly to the first touch of whiskers and adapts to subsequent touch. They also report that there is no positive response to detach. Remarkably, they also found that S1 neurons respond to slip-off almost as strongly as to the first touch. This is a completely new result that suggests that information on slip off is encoded in cortex. Thus, tapering of the whiskers may have functional significance.

Minor issues to address in the revision:

1) The authors should provide more explanations about how they solved [Disp-formula equ8]uations –[Disp-formula equ13]. These equations contain unknown free parameters F, \theta_{obj} and s_{obj}. Presumably these are obtained in the shooting solution? How is stability of solutions obtained with the shooting method?

2) The paper is well written and easy to follow. Clearly, the authors have made a lot of effort in order to make the modeling part accessible to a large audience. Yet, this can be further improved. For instance a sentence like the one opening the second paragraph of the Results (“Generically, the boundary value problem…“) may be very cryptic to a reader without an appropriate mathematical background. Elaborating, somewhat more on the qualitative meaning of a “boundary value problem” or on the meaning of an “unstable solution” should be straightforward and will be of significant help to the reader.

3) The novelty of the theoretical results must be emphasized more in the Discussion. In particular, the authors can elaborate a bit more on the comparison of their approach and results with those of [7].

---

## [Author Response]

We have made textual revisions and a few tweaks to the manuscript with the goal to:

1) Explain in more detail how we solved the boundary-value problem.

2) Explain the mathematical terms for non-mathematicians.

3) Further compare our results to published results in this area based mainly on numerical methods.